# Declines in Physical Activity among New Zealand Adults during the COVID-19 Pandemic: Longitudinal Analyses of Five Data Waves from Pre-Pandemic through April 2021

**DOI:** 10.3390/ijerph19074041

**Published:** 2022-03-29

**Authors:** Oliver W. A. Wilson, Hamish McEwen, Paul Kelly, Justin Richards

**Affiliations:** 1Faculty of Health, Victoria University of Wellington, Wellington 6140, New Zealand; oliver.wilson@vuw.ac.nz; 2Sport New Zealand, Wellington 6140, New Zealand; hamish.mcewen@sportnz.org.nz; 3Physical Activity for Health Research Centre, Institute of Sport, Physical Education and Health Sciences, Moray House School of Education and Sport, University of Edinburgh, Edinburgh EH8 8FF, UK; p.kelly@ed.ac.uk

**Keywords:** exercise, COVID-19, coronavirus

## Abstract

Insights specific to the impact of the COVID-19 pandemic on physical activity participation in New Zealand (NZ) is limited. The purpose of this study was to examine longitudinal changes in leisure-time physical activity from pre-pandemic through April 2021. Demographic characteristics (age, gender, ethnicity, deprivation, disability status) and numerous indicators of leisure-time physical activity participation were assessed in a cohort of NZ adults (n = 1854, 54.6% women) over five waves (pre-pandemic, April 2020, June 2020, September 2020, and April 2021). Repeat measures were completed for: proportion participating in any physical activity; proportion meeting aerobic physical activity recommendations; physical activity duration; number of physical activities. There was a significant (*p* < 0.05) decline in mean physical activity duration and number of activities during the initial lockdown period in April 2020, but the proportion participating in any physical activity and the proportion meeting physical activity recommendations remained stable. However, all four physical activity participation indicators were significantly (*p* < 0.05) below pre-pandemic levels in all subsequent waves. Considerable and sustained declines in NZ adult leisure-time physical activity participation were evident over the first year of the pandemic. As of April 2021, physical activity participation showed limited signs of recovering to pre-pandemic levels.

## 1. Introduction

The COVID-19 pandemic had, and continues to have, a considerable impact on daily life. The timing and responses of countries to the pandemic has varied considerably. Strategies such as lockdowns, social restrictions, and masking mandates impacted the nature of people’s work, the delivery of education, travel methods and freedom, as well as the frequency, intensity, duration, and type of leisure activities people could participate in [1,2,3,4,5]. Although there are variable findings in the international literature, overall results indicate that physical activity levels have been impacted considerably, with decreases observed among children and adults [6,7,8,9,10]. Any deterioration in physical activity is cause for concern, especially considering the association between physical activity and physical and mental well-being [11].

While international evidence provides helpful insight into what may be going on in other countries, limited research has examined the impact of the pandemic on physical activity outcomes in Aotearoa New Zealand (NZ). NZ’s elimination strategy and unique handling of the pandemic compared to most other countries means that people in NZ have experienced fewer lockdowns of shorter duration, but more stringent restrictions during lockdowns [12]. Importantly, border closures aside, the initial success of the elimination strategy has meant that daily life in NZ appeared to resemble “normal” for large portions of the population during the pandemic, particularly during its first year. However, there is growing evidence concerning the disruption the pandemic has caused across multiple sectors in NZ and some data indicating notable changes in the sport and physical activity sector during this period [13,14,15]. For example, injury claim data suggest there was a decrease in participation in team sports, and an increase in activities such as cycling, jogging, and skateboarding [16]. Despite this, the extent to which the pandemic has impacted the physical activity behaviours of people residing in NZ, and in comparison to other countries, remains largely unclear.

The purpose of this study is to examine changes in leisure-time physical activity from pre-pandemic to during the pandemic in NZ. We assessed changes in physical activity at different points during the first year of the pandemic (i.e., April 2020–April 2021) to provide insight into the impact of various pandemic-related public health interventions on physical activity participation levels. Our objective was to understand the extent to which the pandemic has impacted adult physical activity participation in NZ and its trends during this period to inform future interventions aimed at improving participation.

## 2. Materials and Methods

### 2.1. Participants and Recruitment

Data were collected as an extension of the Active NZ Adult Survey, which was paused during the initial response to the pandemic [17]. Pre-pandemic the Active NZ Adult Survey collected data continuously from the beginning of 2017 to the first quarter of 2020. Adults (aged ≥18 years) were recruited to participate in the Active NZ survey using the NZ electoral roll as a sampling frame. Full details of the survey methods are articulated in the annual Active NZ Technical reports [18,19,20]. All pre-pandemic Active NZ survey respondents were asked to provide contact details and indicate whether they agreed to be recontacted for future research purposes. Those people who agreed to be recontacted formed the sampling frame for the cohort study that was established at the beginning of the pandemic. Repeated measures from the participants were collected during four waves after the onset of the pandemic (April 2020, June 2020, September 2020, April 2021). There were 10,174 previous participants who agreed to be contacted and responded to the first follow-up survey. The number of participants with responses across all surveys decreased to 4543 at June 2020, and to 2676 at September 2020.

### 2.2. Measures

The dataset comprises self-reported socio-demographic characteristics and several items about physical activity behaviours. Treatment of data and variables are detailed below.

Age: Participants specified their age in years and were categorised into groups, starting at 18–24 years, followed by 25–34 years, 25–49 years, 50–64 years, 65–74 years, and 75+ years.

Gender: Participants identified their gender (male, female, or gender diverse). Due to limited sample size for gender diverse our analyses focused on cis-gender individuals.

Ethnicity: Participants identified their ethnic group(s) and there was no limit on the number of ethnicities they could choose. For the purposes of these analyses, participants who identified multiple ethnicities were categorised to only one of these in the following order: Māori, Pasifika, Asian, Middle Eastern/Latin American/African (MELAA), European, Other.

Disability status: Participants specified conditions that might affect their physical activity using the Washington Group Short Set of Questions on Disability [21]. For the purposes of these analyses, participants who did not report using a wheelchair, using a walking aid, using prosthetics, or dealing with an ongoing physical illness were classified as someone without a disability.

Deprivation status: Deprivation was determined using the 2013 NZ Index of Deprivation, which combines census data relating to income, home ownership, employment, qualifications, family structure, housing, access to transport and communications to designate small geographic areas (i.e., 60–110 people) with a decile number ranging from 1 (least deprived) to 10 (most deprived) [22]. Participants were classified as residing in low (deciles 1–3), medium (deciles 4–7) and high (deciles 8–10) deprivation areas.

Leisure-time physical activity: There were four indicators for physical activity participation included in the study: (i) Physical activity participation—Participants were asked whether they had participated in any physical activity that was specifically for the purpose of sport, exercise or recreation in the past seven days and the results were dichotomised (yes/no); (ii) Number of activities—Participants were asked to identify what activities they participated in during the past seven days from a list of 59 options (e.g., walking, rugby, other, etc.) and the results were reported for the sum of different activities identified; (iii) Physical activity duration—Participants were asked to self-report the total duration of their participation in physical activities in the past seven days for sport, exercise, or recreation and the results were reported in hours; (iv) Physical activity recommendations-derived by dichotomising physical activity duration according to those who met current aerobic physical activity recommendations (i.e., ≥150 min/week) [11].

### 2.3. Data Analyses

Physical activity varies based on season [23]. As such, to ensure that conclusions were based on changes attributable to the pandemic the physical activity indicators were calibrated for seasonality by determining the relative difference in each month compared to the annual average using the pre-pandemic data collected between 2017–2019. This relative difference was then used to calibrate both the pre- and post-pandemic data for longitudinal comparisons of ‘annualized’ data across different time points. Descriptive statistics were computed to describe the sample. Analyses of categorical variables (i.e., physical activity participation, physical activity recommendations) were conducted using McNemar’s tests. Repeated-measures ANOVA with demographic characteristics (age, gender, ethnicity, socio-economic status, disability status) included as covariates were used to examine changes in continuous variables (i.e., number of activities, physical activity duration). Percentage change relative to pre-pandemic levels was calculated for all of the physical activity indicators at each time point. All analyses used SPSS 28.0 (IBM, Armonk, NY, USA) with significance levels set at *p* < 0.05.

## 3. Results

### 3.1. Participant Characteristics

Participant characteristics descriptive characteristics are displayed in Table 1. A total of 1854 participants provided data at all five time points. Just over half of participants were women. Most participants were aged 50–74 years, European, did not have a physical disability, and resided in low/mid-deprivation areas.

### 3.2. Changes in Physical Activity

Results for the physical activity participation indicators at each time point and longitudinal changes in these are displayed in Table 2.

Physical activity participation. The proportion of people participating in any physical activity in the past week remained stable compared to before the pandemic when the country entered an initial lockdown period (April 2020). This dropped substantially to 6% below pre-pandemic levels when NZ emerged from the initial lockdown (June 2020). It then showed signs of some recovery but was still below pre-pandemic levels in April 2021 when daily life in NZ had largely returned to “normal” (except for border restrictions).

Number of activities. The mean number of physical activities people participated in each week decreased by more than 20% compared to before the pandemic when NZ entered an initial lockdown period (April 2020). It remained around 20% lower through to April 2021.

Physical activity duration. The mean hours of physical activity people participated in each week decreased by 9% compared to pre-pandemic when NZ entered an initial lockdown period (April 2020). It further decreased to 15% below pre-pandemic levels when NZ emerged from the initial lockdown (June 2020) and had not recovered by April 2021.

Physical activity recommendations. The proportion of adults meeting the aerobic physical activity recommendations remained stable compared to before the pandemic when NZ entered an initial lockdown period (April 2020). This then dropped substantially to >7% below pre-pandemic levels when NZ emerged from the initial lockdown (June 2020) and had not recovered by April 2021.

## 4. Discussion

The current study provides insight into changes in leisure-time physical activity participation beyond the first few months of the pandemic. Findings indicate a considerable decline in NZ adult physical activity participation over the first year of the pandemic. However, the magnitude and apparent stability of changes in physical activity participation varied between indicators. Despite some stability during the initial lockdown in the proportion participating in any physical activity and meeting physical activity recommendations, there was a large decline in number of activities and duration of physical activity. As NZ emerged from lockdown in June 2020, all physical activity indicators were lower than pre-pandemic levels. From June 2020 through April 2021, there were variations in how each indicator changed, but the overarching picture is one of large and sustained deficits in physical activity participation when compared to pre-pandemic. This has implications for ongoing physical activity participation, subsequent health outcomes, and relevant intervention planning.

During the initial lockdown, the proportion of adults meeting physical activity recommendations did not change, despite the large decreases in overall physical activity duration. This indicates that the initial large decline in physical activity duration was mainly driven by those who were highly active pre-pandemic. This is consistent with findings from studies that used retrospective data collected about physical activity participation in NZ and is not surprising when considering the severity of the lockdown measures enforced [24,25,26]. The coinciding immediate drop in the number of activities people participated in during the initial lockdown period is somewhat expected given the shutdown of facilities and organized sport that occurred, and the restrictions placed on personal movement within their local communities. The lockdown environment in NZ highlighted the importance of several accessibility barriers previously identified and well established in the physical activity literature [25]. However, the drop in activity number was not accompanied by a change in the proportion of people doing any activity, which indicates people found ways of at least doing some physical activity in their homes and local neighborhoods. From an intervention development perspective, this could be seen as an opportunity to reduce perceived environmental barriers to physical activity and improve utilization of local opportunities (e.g., neighborhood green space and bike-friendly roadways).

However, the subsequent decline in all physical activity indicators when NZ emerged from lockdown in June 2020 suggests that any positive changes in locally accessed physical activity participation was not maintained. As daily life in NZ largely returned to normal, the opportunity to intervene to maintain these behaviors seems to have been missed. This is despite considerable government investment in the initial recovery of play, active recreation, and sport sector [27]. It is possible that the June 2020 data wave was collected before the physical activity sector had restarted its usual operations and as people returned to their previous work routines, which disrupted any new local activity that they had started during the lockdown. In short, we hypothesize that the opportunity to continue new “local” physical activities decreased before the opportunity to recommence previous “organized” physical activity was reinstated with the assistance of unprecedented government financial support.

It is possible that the positive shift in the results for most of the physical activity indicators by September 2020 were signs of recovery towards pre-pandemic behaviors that were supported by the additional funding provided for the sector. However, these changes were small, and all indicators remained significantly below pre-pandemic levels. Most of the indicators then deteriorated to further below pre-pandemic levels by April 2021, despite minimal social restrictions being in place for most of the time across the country for over nine months. This is contrary to previous reports that concluded physical activity levels in NZ had been maintained over the longer term [25]. However, these previously published findings were based on results from retrospective reporting of pre-pandemic physical activity levels from a relatively small sample recruited on voluntary basis through social media. Our study has A substantially lower risk of measurement and sample bias, which gives us more confidence in the results we have reported.

Our findings are also consistent with global trends of a negative impact of the pandemic on adult physical activity participation both in the short and longer term [6]. There are already numerous studies on this topic and the number is likely to continue growing. Findings from nationally representative samples may be considered more informative than those from highly selective or biased samples and provide a good point of reference for our results. In January 2022, Strain et al. [7] looked at international evidence from studies that attempted to recruit nationally representative samples using a semi-systematic approach to reviewing the literature (searches conducted in June 2021). They identified seven peer-reviewed articles that reported how physical activity levels had changed before, during and after the introduction of COVID-19 restrictions. These studies came from Belgium, France, Germany, Saudi Arabia, the United Kingdom, and the United States. While the findings varied across different population groups, the overall trend was for decreasing physical activity levels [7].

The magnitude and apparent permanency of the unfavorable change in what could now be considered habitual physical activity is cause for serious concern. Particularly given physical activity at a population level is typically relatively stable over time, except for minor seasonal fluctuations [28]. The implications of not acting upon the confluence of the pandemic and the physical inactivity that existed prior to the emergence of COVID-19 have been laid bare [29]. Our findings highlight the need for action now, more than ever, at a population level to increase the physical activity of all New Zealanders. The NZ government initially announced considerable investment ($264 million) into the recovery of the play, active recreation, and sport sector. Over 80% of this funding has already been allocated and our results indicate that the recovery from the COVID pandemic is far from complete. A substantial proportion of this funding has gone into keeping pre-COVID structures and stakeholders afloat, including substantial investment to support jobs and industry across the physical activity and high-performance sport sector. This has left a smaller investment pool to address the extended impacts of the pandemic on population physical activity participation, which may require innovative ways of promoting physical activity in a fundamentally different world. To guide this investment, further research to understand the cause of reduced physical activity participation and identifying key population groups whose activity levels have been affected most by the pandemic is indicated.

Our study is not without limitations. Firstly, while the longitudinal design of this study means data are less susceptible to recall bias compared to much of the international evidence [6], it is widely accepted that self-reported leisure-time physical activity measures are less valid in comparison to objective measurement devices of bodily movement such as accelerometers [30,31]. Despite this, the use of multiple validated and consistent self-report measures of physical activity over time strengthened the conclusions we could draw about the behavioral impacts of the pandemic. Secondly, our measures only included participation in leisure-time physical activity and did not capture changes in active transport. Changes in working practices during the pandemic, such as increasing numbers of people working from home, are likely to have impacted physical activity in the transport domain. Further research is warranted to assess the magnitude of these changes and its impact on overall physical activity levels (i.e., was it displaced by physical activity in other domains). Thirdly, although the origins of our sample frame were from the electoral role, sensitivity analysis revealed that our cohort were more physically active than the general population according to their baseline Active NZ results.

There was also considerable participant attrition with each wave of data collection. Admittedly, this is evidence of a selection bias in our sample, and by virtue of being more active pre-pandemic, the cohort did have further to fall. However, it also suggests that our study comprised a sample that may be more likely to re-engage in pre-pandemic physical activity opportunities and the ongoing deficits may be even more pronounced in the broader population. Finally, exclusively focusing on adults prevents the generalization of findings to adolescents and children. This is particularly pertinent given recently published findings indicating that the impact of the pandemic on physical activity differed between children and adults in Australia, where the pandemic was initially handled similarly [32].

Despite these limitations, our study provides the most robust methods yet for describing how leisure-time physical activity participation among adults in NZ changed during the first year of the pandemic. It also extends most previous global research by analyzing individually linked data collected pre-pandemic (i.e., not retrospectively) to data collected using the same survey items at multiple stages during the pandemic. Finally, although the sample size was not adequate to explore inequities across different socio-demographic groups, the sample size was large enough to monitor changes in the cohort and confidently draw conclusion about changing physical activity behaviors in NZ.

## 5. Conclusions

In summary, the leisure-time physical activity participation of NZ adults remained well below pre-pandemic levels a little over a year into the pandemic despite relatively few social limitations. Our results present important insights that might help to inform public health policy and strategies as the current pandemic continues to evolve and in future pandemic situations. Specifically, the apparent change in habitual physical activity, especially given it is observed among a relatively active sample, is cause for serious concern. Urgent action is needed to re-activate NZ adults and prevent the unfavorable physical and mental health outcomes likely to result if the trends observed persist in the long-term at a broader population level [11].

## Figures and Tables

**Table 1 ijerph-19-04041-t001:** Participant characteristics (*n* = 1854).

	*n*	%
Age		
18–24	102	5.5
25–34	118	6.4
35–49	356	19.2
50–64	600	32.4
65–74	479	25.8
75+	199	10.7
Gender		
Men	841	45.4
Women	1013	54.6
Ethnicity		
European	1634	88.1
Maori	127	6.9
Pasifika	13	0.7
Asian	58	3.1
MELAA	11	0.6
Other	11	0.6
Disability status		
Without disability	1466	79.1
With disability	388	20.9
Deprivation status		
Low-deprivation	718	38.7
Mid-deprivation	757	40.8
High-deprivation	379	20.4

**Table 2 ijerph-19-04041-t002:** Changes in adult physical activity during the first year of the COVID-19 pandemic in New Zealand.

	Participate in Physical Activity (Yes/No)		Number of Activities (per Week)		Physical Activity Duration (Hours/per Week)		Meet Recommendations (≥150 min/Week)	
	%	PP diff (%)	Mean (SE)	PP diff (%)	Mean (SE)	PP diff (%)	%	PP diff (%)
PP ^#^	83.7		2.62 (0.04)		6.80 (0.15)		73.6	
April-2020 ^#^	84.8	1.3	2.02 (0.04)	−22.9 *	6.16 (0.13)	−9.4 *	72.6	−1.4
June-2020 ^#^	78.9	−5.7 *	2.07 (0.04)	−21.1 *	5.81 (0.12)	−14.6 *	68.2	−7.3 *
September-2020 ^#^	79.9	−4.5 *	2.14 (0.04)	−18.3 *	5.99 (0.13)	−12.0 *	69.1	−6.2 *
April-2021 ^#^	80.8	−3.4 *	2.04 (0.04)	−22.2 *	5.74 (0.14)	−15.6 *	67.6	−8.3 *

* = < 0.05; All physical activity variables differed significant from pre-pandemic at each recontact time point; Tests for statistical significance were adjusted for gender, age, ethnicity and disability status, deprivation status; PP = Pre-pandemic. ^#^ Physical activity indicators measured pre-pandemic and in all subsequent waves were calibrated based on the month the data was collected to create an annualized estimate adjusted for seasonality.

## Data Availability

Publicly available datasets were analyzed in this study. This data can be provided on request from research@sportnz.org.nz.

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
