# Peer review of "Declines in Physical Activity among New Zealand Adults during the COVID-19 Pandemic: Longitudinal Analyses of Five Data Waves from Pre-Pandemic through April 2021"

_ijerph, 2022, doi:10.3390/ijerph19074041_

Round 1

Reviewer 1 Report

The main concern of this study is lack of novelty (the data was almost one year ago) and generalizability (only for New Zealand adults). In addition, the result is totally expectable. There was no novel technique to tract the physical activity (e.g., accelerometer)

Author Response

Dear editors and reviewers,

Thank you for your helpful and informative reviews. We have addressed each of your comments as track changes in the revised manuscript and outlined our response in the table below.

Please let us know if you have any further concerns with how we have addressed the reviewer comments.

Thank you again for considering our paper for publication in the International Journal of Environmental Research and Public Health.

Kind regards,

Oliver Wilson (on behalf of the co-authors)

Reviewer 1

The main concern of this study is lack of novelty (the data was almost one year ago) and generalizability (only for New Zealand adults). In addition, the result is totally expectable. There was no novel technique to tract the physical activity (e.g., accelerometer)

Thank you for reviewing our manuscript and raising these important questions

Though data were collected almost a year ago, very few studies have followed a cohort of participants for such a length of time during the pandemic. A longitudinal study, such as ours, adds to the existing evidence that has mostly applied weaker cross-sectional or retrospective study designs. We detail the shortcomings of prior NZ research in particular in lines 225 through 230 of the manuscript.

It is also notable that not all of the previous international studies on the impact of Covid on physical activity participation have concluded with the same changes in physical activity. We have more clearly articulated in this our introduction (lines 35-37) and believe this clearly indicates that the results were not necessarily expectable, but warranted investigation.

With respect to generalizability, no one country has experienced the pandemic in the same manner. Thus, generalizability is limitation not unique to our study. Moreover, unlike prior research we calibrated physical activity variables to control for the natural seasonal fluctuations observed in physical behaviours. Doing so arguably makes the findings of our study, particularly those pertaining to the impact of the pandemic rather than the season, more comparable with others who elect to adopt a similar approach

As far as novelty, we simply took advantage of a unique event in the pandemic to begin to collect longitudinal data. We contest that the pandemic provides novelty in and of itself. Specifically, in the time since physical activity has become a recognized field of research there has never been a global event that has disrupted normal behaviours so much. Furthermore, the measures used were predetermined by the data that was available to us prior to the pandemic. In any case, given the behavioural impact of the pandemic we believe that self-report measures of physical activity behaviour are an appropriate measure to use in our study, rather than accelerometers that actually measure movement rather than physical activity. Specifically, self-report measures enable the assessment of different components of physical activity, such as the number of activities participated in, that are not captured with device based-measures. Being able to examine all of these components simultaneously whilst calibrating for seasonal differences in behaviours makes our study relatively unique.

Reviewer 2 Report

Dear Authors,

Thank you for your manuscript. The manuscript has a relevant topic for public health and sports sciences. It is well written and easy to read. The aim of the longitudinal study was to assess the impact of the COVID19 pandemic on physical activity participation in New Zealand. Findings of this longitudinal study are important globally reporting physical activity changes in a representative sample in four waves comparing pre-pandemic and pandemic situations.

There are some comments that I believe might increase the quality of the manuscript.

Gender, mean age of the sample and age range of participants in each wave might be included in the abstract.

I found no information reporting numbers of respondents in all four waves of research. Was it 1854 in all waves? Please include information on how many participants gave permission to contact them for future studies, how many filled questionnaires in the first, second and other waves of research. 
In participants' characteristics, it is reported that most of the participants were women, I would disagree that 54.6 % of women is a major part.

Practical implications of the study might be discussed at the end of the Discussion section. The study presents important insights that might help to inform public health policy and strategies in future pandemic situations.

All the best in your further research!

Author Response

Dear reviewers,

Thank you for your helpful and informative reviews. We have addressed each of your comments as track changes in the revised manuscript and outlined our response in the table below.

Please let us know if you have any further concerns with how we have addressed the reviewer comments.

Thank you again for considering our paper for publication in the International Journal of Environmental Research and Public Health.

Kind regards,

Oliver Wilson (on behalf of the co-authors)

Reviewer 2

Thank you for your manuscript. The manuscript has a relevant topic for public health and sports sciences. It is well written and easy to read. The aim of the longitudinal study was to assess the impact of the COVID19 pandemic on physical activity participation in New Zealand. Findings of this longitudinal study are important globally reporting physical activity changes in a representative sample in four waves comparing pre-pandemic and pandemic situations.

There are some comments that I believe might increase the quality of the manuscript.

Thank you for taking the time to provide such well-considered feedback on our manuscript

Gender, mean age of the sample and age range of participants in each wave might be included in the abstract

Thank you for these suggestions. We have now added gender to the abstract. We cannot provide a mean or range for age, as the data collected categorized age into groups as reported in Table 1. We have not provided demographic characteristics of the sample at each wave, because the participants included in the analysis were consistent across all waves and are therefore consistent at all time point.

I found no information reporting numbers of respondents in all four waves of research. Was it 1854 in all waves? Please include information on how many participants gave permission to contact them for future studies, how many filled questionnaires in the first, second and other waves of research. 

Thank you for raising this important issue in our analytical approach. We had 1854 participants respond across all time points. We have included the sentence below at line 74 within the manuscript

‘There were 10,174 previous participants who agreed to be contacted and responded to the first follow-up survey. The number of participants with responses across all surveys decreased to 4,543 at June 2020, and to 2,676 at September 2020.’

We acknowledge this as a limitation of our study and discuss its implications in Line 273-281

In participants' characteristics, it is reported that most of the participants were women, I would disagree that 54.6 % of women is a major part.

Thank you for identifying this oversight, we have reworded this sentence. Line 134-135

Practical implications of the study might be discussed at the end of the Discussion section. The study presents important insights that might help to inform public health policy and strategies in future pandemic situations

Thank you for your comment. We have dedicated an entire paragraph to the key practical implications of our findings in the discussion (line 243-260). However, we acknowledge that we could have made more of these implications in the conclusion. We have added text to acknowledged that the study presents important insights that might help to inform public health policy and strategies in our conclusion (line 297-299)

Reviewer 3 Report

This is a great paper and very well written with information that should be shown to many. However, why did the authors not choose to use the Leisure time physical activity questionnaire or the Global physical activity questionnaire?

You address in the discussion limitations of utilizing self-report, but it seems as though you created your own questionnaire which do not (correct me if you have the data) have validity measures with accelerometers. Please clear this up and then I believe that this should be accepted. 

Author Response

Dear editors and reviewers,

Thank you for your helpful and informative reviews. We have addressed each of your comments as track changes in the revised manuscript and outlined our response in the table below.

Please let us know if you have any further concerns with how we have addressed the reviewer comments.

Thank you again for considering our paper for publication in the International Journal of Environmental Research and Public Health.

Kind regards,

Oliver Wilson (on behalf of the co-authors)

Reviewer 3

This is a great paper and very well written with information that should be shown to many.

Thank you, we appreciate the feedback you offered that helped us to further improve our manuscript

However, why did the authors not choose to use the Leisure time physical activity questionnaire or the Global physical activity questionnaire?

Thank you for raising this important point. The questionnaire we used throughout the study was shaped by the previous application of these survey items prior to the pandemic in NZ. The survey items used were based on internationally validated measurement tools for assessing physical activity that had been previously tested and adapted for application in the NZ context. Using the same survey items throughout our study was a pragmatic decision that ensured we had continuity in the measurement tool, which has been shown to be critical for generating comparable estimates of self-reported physical activity behavior.

You address in the discussion limitations of utilizing self-report, but it seems as though you created your own questionnaire which do not (correct me if you have the data) have validity measures with accelerometers. Please clear this up and then I believe that this should be accepted. 

The self-report measures used in our study underwent extensive face validation when they were first adapted for use in NZ. They have subsequently undergone extensive validation against the International Physical Activity Questionnaire and the widely used Physical Activity Single Item Questionnaire. The results demonstrate that the measure used in our study is a robust and valid measure of leisure time physical activity (Ref: Bauman, A. Measures of physical activity in the Active New Zealand surveys, 2021. Sport NZ: Wellington).

Reviewer 4 Report

Review

Declines in physical activity among New Zealand adults during the COVID-19 pandemic: Longitudinal analyses of five data waves from pre-pandemic through April 2021

Thank you for the opportunity to review this manuscript regarding physical activity change during COVID-19.

Physical activity has a profound implication for many areas of life and well-being and thus the topic and the aim of the research is important for public health in response to COVID-19 issues.

The strength of the study in its longitudinal approach to consider changes in physical activity. However, the weakness of the study is in limited physical activity measurements (no type of activity or intensity provided). Authors discussed and acknowledged these limitations.

Below are my comments that I believe can improve the quality of the manuscript.

Line 35, There are also published studies (including in Int. J. Environ. Res. Public Health) whish showed that physical activity increased or maintained during the pandemic. Please make a balanced introduction.

Table 2, please make the order of the column, consistent with the order in the method section (e.g, “participate in physical activity” was the first variable described in method).

How the percentages (diff, %) were calculated in Table 2. Because % numbers are quite big, eg -22.9%. Could be reasonable to indicate in data analysis section how diff, % were calculated.

It seems that 73 % meet the PA recommendations for 150 min a week. I think it is rather an active slice of population. Could be useful to say about this in discussion.

Line, 176. “During the initial lockdown the distribution …detrimental to overall health recommendations ”. Consider revising, as the meaning a bit obscured. Recommendations can’t be harmed.

Line 178, “Specifically, the proportion of adults meeting physical activity recommendations 178 did not change and…”. Is it? In table 2 the proportion declines, than kind of stabilizes.

Line 180-186. There are recently published papers in the special issue of this journal about changes in types of physical activity etc. I think they are relevant in the context of the discussion and your findings. Unfortunately, I can’t provide a direct reference due to possible conflict of interests. Please consider adding most recent findings in the discussion.

What was the nature and severity of COVID restrictions in NZ in comparison with other countries? Is this relevant to the findings?

Author Response

Dear reviewer,

Thank you for your helpful and informative reviews. We have addressed each of your comments as track changes in the revised manuscript and outlined our response in the table below.

Please let us know if you have any further concerns with how we have addressed the reviewer comments.

Thank you again for considering our paper for publication in the International Journal of Environmental Research and Public Health.

Kind regards,

Oliver Wilson (on behalf of the co-authors)

Reviewer 4

Thank you for the opportunity to review this manuscript regarding physical activity change during COVID-19.

Physical activity has a profound implication for many areas of life and well-being and thus the topic and the aim of the research is important for public health in response to COVID-19 issues.

The strength of the study in its longitudinal approach to consider changes in physical activity. However, the weakness of the study is in limited physical activity measurements (no type of activity or intensity provided). Authors discussed and acknowledged these limitations.

Thank you for your feedback which has helped us to enhance our manuscript.

Line 35, There are also published studies (including in Int. J. Environ. Res. Public Health) which showed that physical activity increased or maintained during the pandemic. Please make a balanced introduction.

Thank you for identifying this. We have acknowledged there are variable findings in our introduction and added relevant references in line 35-37.

Table 2, please make the order of the column, consistent with the order in the method section (e.g, “participate in physical activity” was the first variable described in method).

We appreciate you picking this up. Table 2, and the accompanying next have been revised accordingly so that the order is consistent with the methods

How the percentages (diff, %) were calculated in Table 2. Because % numbers are quite big, eg -22.9%. Could be reasonable to indicate in data analysis section how diff, % were calculated.

Thank you for this suggestion. We have added additional detail to the manuscript (line 128-129)

It seems that 73 % meet the PA recommendations for 150 min a week. I think it is rather an active slice of population. Could be useful to say about this in discussion.

Thank you for identifying this issue. We agree with your observation and have addressed in the discussion (lines 273-281).

Line, 176. “During the initial lockdown the distribution …detrimental to overall health recommendations ”. Consider revising, as the meaning a bit obscured. Recommendations can’t be harmed.

Thank you for identifying this, we have removed the confusing statement and reworded the subsequent sentence.

Line 178, “Specifically, the proportion of adults meeting physical activity recommendations 178 did not change and…”. Is it? In table 2 the proportion declines, than kind of stabilizes.

Thank you for picking up the confusing way we have presented this finding in the discussion. In responding to your comment above, we have clarified we are referring to the initial part of the pandemic (line192-193).

Line 180-186. There are recently published papers in the special issue of this journal about changes in types of physical activity etc. I think they are relevant in the context of the discussion and your findings. Unfortunately, I can’t provide a direct reference due to possible conflict of interests. Please consider adding most recent findings in the discussion.

Thank you for the directing us to this recent special addition. We believe we have added the relevant citations to the manuscript, but we would be happy to consider other suggestions if we have missed anything in our updated literature search.

What was the nature and severity of COVID restrictions in NZ in comparison with other countries? Is this relevant to the findings?

Thank you for recognizing this important consideration that we agree has clear implications on our findings. The nature and severity of NZ’s covid response strategy ranged considerably over the course of period of this study. We provide some background in lines 42-47, and context when reporting results in lines 142-148. We also provide an indication of the severity of the lockdown severity and “normality of life” as we consider the implications of our results throughout the discussion (line 197-198, 212-214, 226-228). 

Round 2

Reviewer 1 Report

Thank you for detailed explanation.